# A Model Calibration Approach to U-Value Measurements with Thermography

**Dhruvkumar Patel [1], Jacob Estevam Schmiedt [1,*], Marc Röger [2] and Bernhard Hoffschmidt [3]**

[1] Deutsches Zentrum für Luft- und Raumfahrt e.V. (German Aerospace Center), Institute of Solar Research, Karl-Heinz-Beckurts-Str. 13, 52428 Jülich, Germany; dhruvkumar.patel@dlr.de

[2] Deutsches Zentrum für Luft- und Raumfahrt e.V. (German Aerospace Center), Institute of Solar Research, Paseo de Almería, 04001 Almería, Spain; marc.roeger@dlr.de

[3] Deutsches Zentrum für Luft- und Raumfahrt e.V. (German Aerospace Center), Institute of Solar Research, Linder Höhe, 51147 Köln, Germany; bernhard.hoffschmidt@dlr.de

* Correspondence: jacob.estevamschmiedt@dlr.de

**Abstract:** The thermal properties of a building envelope are key indicators of the energy performance of the building. Therefore, methods are needed to determine quantities like the thermal transmittance (U-value) or heat capacitance in a fast, reliable way and with as little impact on the use of the building as possible. In this paper a technique is proposed that relies on a simplified electrical analogical model of building envelope components which can cover their dynamic thermal behavior. The parameters of this model are optimized to produce the best fit between simulated and measured outside surface temperatures. As the temperatures can be measured remotely with an infrared camera this approach requires significantly less installation effort and intrusion in the building than other methods. At the same time, a single measurement provides data for a large range of locations on a facade or a roof. The paper describes the method and a first experimental implementation of it. The experiment indicates that this method has the potential to produce results which have an accuracy that is comparable to standardized reference methods.

**Keywords:** U-value; thermal transmittance; heat capacitance; thermography; model calibration; building energy performance

## 1. Introduction

The high share of energy consumption in the building sector is a global concern. In the EU private households alone were responsible for 27% of the final energy consumption in 2020. More than 60% of this energy is used for space heating and only 27% of this energy comes from renewables and biofuels [1]. Similarly, the International Energy Agency states that the buildings and construction sector account for 36% of the global final energy consumption in 2019, including energy for manufacturing of building materials, and that space heating contributed still the largest share of the energy consumption in operation [2]. A reduction of energy demand for heating and cooling will, therefore, help in curbing the greenhouse gas emissions. In general, there are three major factors that determine the energy performance of a building: The behaviour of the users, the efficiency of the HVAC system, and the thermophysical properties of the building envelope. A key indicator for the latter one is the thermal transmittance (commonly referred to as the "U-value") that is given as the heat flux through an envelope element of unit area at a difference of 1 K between inside and outside air temperatures. It has been observed frequently that design values which are based on information from manufacturers of building materials or standard assumptions and thermal transmittances measured on real buildings differ significantly. E.g., Li et al. conclude that "the mean U-value of English solid-walled properties is significantly lower than the CIBSE Guide A value" and that "the distribution of U-values is so large that the on-going use of a single mean cannot be justified when

assessing individual properties" [3]. Xeni et al. find deviations of up to 30% between measured transmittances and theoretical design values [4]. In addition, for many old buildings the design values are not known. The deviation between assumed and real thermal transmittance has been identified by various authors as one of the important contributions to the so-called energy performance gap [5] between simulation and real energy use for domestic [6] and non-domestic buildings [7]. So to provide a solid basis for decisions about possible retrofit options, fast and accurate measurement methods are required to measure real thermal transmittances of building envelopes. The heat capacitance of building envelopes is another relevant quantity for its energy performance and the user comfort as it can flatten peaks in heating and cooling demand because, e.g., heat can be stored in the building materials and it can be released to the room when the outside temperature is extremely low for a short time [8].

There are standardized methods to determine the thermal transmittances of building envelope elements. These rely on averaging procedures to approximate steady-state conditions and, therefore, require relatively long measurement times of several days. The most common one is the heat flux meter (HFM) method [9] which uses air temperature sensors and heat flux meters that are attached to the building element. Therefore, they only capture the heat flow through a small area and heterogeneities are not detected.

Infrared (IR) thermography is a well known tool to qualitatively assess the insulation quality of building envelops and to find heat bridges. It is also used to identify representative locations for HFM measurements [10]. For many years researchers are working towards a quantitative evaluation of the thermal performance of building envelopes using thermography [11]. The general applicability of IR thermography for transmittance measurements under steady-state conditions has been demonstrated in the laboratory, e.g., by Simões et al. [12]. In 2017 Tejedor et al. described a method that uses a time series of internal IR measurements and the steady-state assumption to determine transmittances [13]. In 2018 a procedure was standardized that is performed from the inside of a building and also attempts to approximate a steady-state measurement [14]. Such approaches can cover larger areas than HFM measurements and inform about inhomogeneitis [15] but demand nearly constant environmental conditions, so that the measurements usually must be repeated several times [14]. Tejedor et al. propose to use statistical tests or signal modelling to reduce the necessary measurement duration [16]. However, the applicability of their analysis is limited to measurements under external conditions that lead to a steady-state heat flow through the building envelope. Attempts for quantitative external thermography with the steady-state assumption can also be found in the literature. Fokaides and Kalogirou find that thermographic measurements for five dwellings in Cyprus deviate between 5% and 10% from standard assumptions for masonry [17]. Mahmoodazadeh et al. stress the possibility that IR thermography is better suited used to assess inhomogeneous building envelopes [18]. Patel et al. perform a detailed uncertainty analysis of the steady-state approach for external thermographic measurements of the thermal transmittance. They stress that the measurement of thermal radiation that is received by the measured object from the surrounding is very important for accurate thermal transmittance measurements [19].

Model calibration is an approach that does not rely on the steady-state assumption and, therefore, is expected to give higher accuracy and allow for shorter measurement times than the standardized methods [20]. In order to capture also the dynamic behaviour of the heat transport through the building, envelope models may contain also thermal capacitor or thermal mass (TM) elements. Thus, model calibration is also able to provide information about heat capacities. In literature, several approaches are found that use time series of sensor data for energy consumption and other quantities together with weather data to determine an overall heat transfer coefficient for the complete building envelope: Braun and Chaturvedi argue that models with a simple physical representation of energy flows in the building can be calibrated to predict the heat load of buildings for control and diagnosis purposes [21]. Similarly, O'Neill et al. propose to use a electrical circuit-equivalent model of two capacitors and four resistors to estimate the thermal load of medium-sized commercial

building [22], and Rouchier et al. propose how such a calibration can be performed on-line during the operation of a building [23]. While these approaches may be useful for control purposes or to assess the overall energy performance of a building, they can hardly identify specific weaknesses in parts of the building energy system like the building envelope.

Other approaches use heat flux meter data to identify the local thermal transmittance or thermal resistance at the respective measurement spot [20,24]. Arregi et al. compare a steady-state approach with a lumped resistance-capacitance that contains a single capacitor and a distributed capacitance model. The models are calibrated using measured heat fluxes through a simple concrete wall and the authors find that the estimation of thermal resistance is less sensitive to variations in the external conditions if a dynamic model is used. However, the lumped resistance-capacitance model showed larger deviations from the data when fast changes in the temperatue appeared and the capacitance was less consistent for varying calibration intervals than for the distribute capacitance model [25]. This indicates that also the steady-state approaches for thermographic transmittance measurements from the outside are sensitive to external conditions and require either long measurement times for averaging or very restrictive conditions on the environmental conditions that must be fulfilled to use the measurement data.

In this paper we present a model calibration approach to calculate thermal transmittances from time series of IR images that does not rely on the steady-state assumption. As we use image data, the method has the potential to give thermal transmittances for large portions of the building envelope that cover several components from a single series of measurements. At the same time, they allow for a spatial resolution of the thermal transmittances that is only limited by the resolution of the IR sensor in the camera. Therefore, it can also provide information about the homogeneity of the building envelope from a single series of measurements. In this paper we describe the general methodology of the approach for a wall element and a first application to a real wall under simplified environmental conditions. In Section 5 we mention what modifications to the approach will be necessary to apply it for other building envelope components.

## 2. Method

We use a model calibration approach to determine the thermal transmittance that optimizes the model parameters to reproduce time series of surface temperatures which were measured by IR thermography. For this, the wall element is described by a physically informed lumped R-C model [26] as a series of three resistances (R) and two capacitors (C) and cover the dynamics of the heat transfer. In the first two subsections of this section the model details and the calibration procedure are described. In the third subsection we describe the experimental setup that was used to test our method on a real wall. The method is described here for a wall element. Necessary adaptations for other types of building envelop elements are discussed in Section 5.

### 2.1. Envelope Model

The system that our model has to represent is visualized in Figure 1 for the example of a wall element. In the R-C model the wall is modelled as a series of thermal resistors and TM. The internal and external boundary conditions are calculated from measured time series of environmental parameters.

We build the model using the open source modelling language Modelica. The models for the individual resistor and capacitor elements of the wall are taken from the Modelica Standard Library [27]. The visualization by the OpenModelica Connection Editor [28] of a model with two TM and three resistors (2TM model) is shown in Figure 2. It represents a set of differential equations that approximate a one-dimensional heat flow through the wall. The boundary conditions which are required to solve the differential equations are determined by the heat exchange with the internal and external environment through convection, absorption, and emission of radiation.

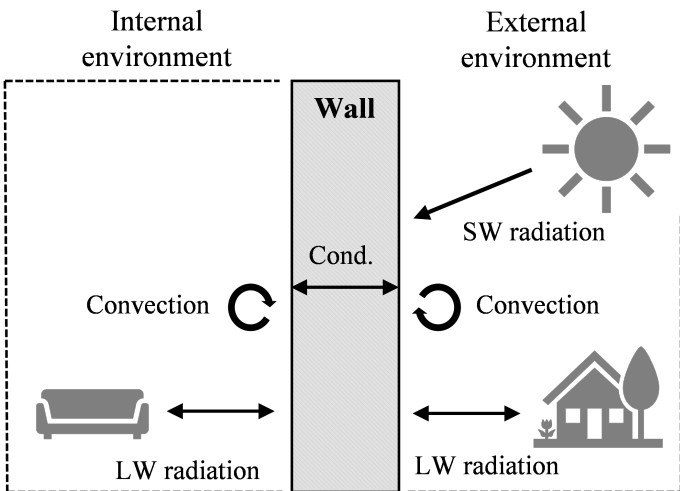

**Figure 1.** Sketch of processes that determine the heat flow through a wall element. Heat is transferred through the wall by thermal conduction. It exchanges heat with the wall by convective heat transfer to the air and long-wave (LW) radiation exchange with objects in its surroundings. In this figure these objects are represented by the sketch of a sofa on the internal side and a sketch of a building and plants on the external side of the wall. On the external side, additional thermal energy is supplied to the wall by the absorption of short-wave (SW) radiation from the sun.

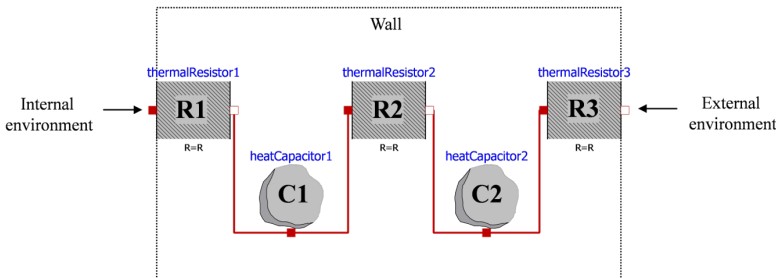

**Figure 2.** 2TM wall model without the boundary conditions. The visualization was generated using the Openmodelica Connection Editor [28].

A thermal resistor element stands for

$$q = \frac{1}{R_{\text{therm}} A} (T_1 - T_r),$$

where $q$ is the heat flux, $R_{\text{therm}}$ is the thermal resistance of the element, $A$ its area, and $T_{l,r}$ are the temperatures on the right and left side of the resistor element, respectively. The capacitor or TM elements stand for

$$C\frac{dT}{dt} = qA,$$

where $C$ is the heat capacity and $\frac{dT}{dt}$ is the derivative of $T$ with respect to time $t$. The connections with red lines in Figure 2 mean that the temperatures at the connected points are set equal and that the heat flux into one element equals the heat flux out of the other element. If one element is connected to more than a single other element it means that the heat fluxes of those other elements are summed up.

In priciple, an arbitrary number of alternating resistor and TM elements can be used. In general, the different TM elements do not necessarily represent different material layers but arise from a discretization of the distance between inner and outer surface. Smaller numbers of elements lead to a smaller parameter space which reduces the time of the

calibration process. Larger numbers give more realistic representations of the temperature profile in the wall and, especially, reproduce fast surface effects better that can appear, e.g., due to changing direct solar irradiation. Models with two TM and three resistors have turned out to be a good trade-off in a series of experiments that we ran and are frequently used in the literature [20,29,30]. If the wall is known to be approximately symmetric the number of parameters can be further reduced by setting equal the resistances of the outer and inner resistor elements and the thermal capacities of the two TM. This assumption reduces the computation time but is not necessary for the applicability of the method.

The boundary conditions are determined by the internal and external environmental conditions that are illustrated in Figure 1. There are three processes that are considered here and add up as contributions to the boundary conditions. One is convective heat exchange between the wall and the surrounding air. Another one is the exchange of thermal long-wave (LW) radiation between the wall and surrounding surfaces. And the third one is the absorption of solar short-wave (SW) radiation. In our case we only consider short-wave radiation coming from the sun. The convective term is given by

$$q_{\text{conv}} = h(T_{\text{wall}} - T_{\text{air}}), \tag{1}$$

where $h$ is the convective heat transfer coefficient, $T_{\text{wall}}$ is the surface temperature of the wall and $T_{\text{air}}$ is the air temperature. The long-wave radiation term is given by

$$q_{\text{lw}} = \varepsilon\sigma\left(T_{\text{wall}}^4 - T_{\text{surr}}^4\right), \tag{2}$$

where $\varepsilon$ is the emittance of the wall surface in the thermal infrared range, $\sigma$ is the Stefan–Boltzmann constant, and $T_{\text{surr}}$ is a weighted average over the radiation temperatures of the surrounding objects. The radiation temperature of an object is understood here as the temperature that a black body which emits the same intensity of thermal radiation (integrated over the observed range from 7.5 to 14 μm) would have.

The short-wave radiation term is given by

$$q_{\text{sw}} = \alpha I_{\text{GTI}}, \tag{3}$$

where $\alpha$ is the absorptance of the wall surface in the short-wave range and $I_{\text{GTI}}$ is the global tilted solar irradiance that hits the wall surface and depends on the position of the sun and the orientation of the wall.

We assume a convective term (Equation (1)) and a long-wave term (Equation (2)) on the inner and outer side of the wall model, while the short-wave term (Equation (3)) appears only for the outer side. In order to solve the differential equations, the minimum required knowledge to provide these boundary conditions are the inside and outside air temperature, an average radiation temperature of the surrounding objects, and the global tilted solar irradiance which can be measured directly or calculated from other solar irradiance measurements.

The whole system of ordinary differential equations with boundary conditions is implemented in Modelica. Given the values of all parameters and the external conditions it can be solved to obtain heat fluxes and temperatures for each element at every moment in time. We choose literature values for for the emittance, the short wave absorptance, and the indoor convective heat transfer coefficient $h_{\text{in}}$. On the outside the choice of $h$ depends on the wind conditions. In a protected spot with no or almost no wind the convective heat transfer coefficient can be calculated using the Nusselt number Nu that depends on $T_{\text{wall}}$ and $T_{\text{air}}$ [31] as

$$h_{\text{out}} = \frac{\text{Nu} \cdot k}{L},$$

where $k$ is the thermal conductivity of the wall and $L$ is its height. If the wind speed implies forced convection an appropriate model must be chosen according to the site-specific conditions. For many situations the model by Liu and Harris [32] provides a good trade-

off between accuracy and simplicity. The thermal transmittance is obtained from these quantities as

$$U = \frac{1}{h_{\text{in}}^{-1} + A \sum_i R_{\text{therm},i} + h_{\text{out}}^{-1}}.$$

### 2.2. Model Calibration

While the external and internal environment conditions can be determined directly by measurements, the thermal resistances and capacities of the model elements that represent the bulk wall cannot be accessed by a direct measurement. However, these are particularly interesting because they give a major contribution to the thermal transmittance of the wall and are responsible for its thermal storage properties. These are determined by the model calibration process. We record a time series of surface temperatures using an IR camera together with the environmental conditions that determine the boundary conditions. Then we optimize the unknown resistances and capacities of the model, so that the surface temperature predicted by the model, given the recorded environmental conditions, and the measured time series of surface temperatures fit as well as possible. We have implemented the optimization in a Python script that interfaces with the Modelica model through OMPython [28]. We use the Levenberg-Marquardt method that is implemented in the LMFIT package [33] to minimize the objective function

$$\chi^2(\mathbf{R}, \mathbf{C}) = \sum_{i=1}^{N} \left( \frac{T_{\text{wall},i}^{\text{meas}} - T_{\text{wall},i}^{\text{model}}(\mathbf{R}, \mathbf{C})}{\sigma_i} \right)^2,$$

where $N$ is the number of time steps for that measurements were taken, $T_{\text{wall},i}^{\text{meas}}$ and $T_{\text{wall},i}^{\text{model}}$ are the measured and predicted temperatures at time step $i$, respectively, $\sigma_i$ is the estimated uncertainty of the measured temperature at time step $i$, $\mathbf{R}$ is a vector that contains the resistance values of the model elements, and $\mathbf{C}$ is a vector that contains the capacity values of the model elements.

To assess the quality of the fit we also calculate the reduced chi-square after the optimization as

$$\chi_r^2 = \frac{\chi^2}{N - n_p},$$

where $n_p$ is the number of optimized parameters. The total thermal resistance for an element of a location is the sum over the resistances of the resistor elements in the model.

### 2.3. Experimental Setup

The experimental test of the proposed method is performed on a small test building which is shown in Figure 3a. The wall that we used for the experiment is made of pumiced stone with light weight concrete and has an area of $A = 8\,\text{m}^2$. It is a plastered single-layer wall and it has a thermal transmittance of $U_{\text{man}} = 1.21\,\text{W/m}^2\text{K}$ according to the manufacturer. The building was heated with a portable 9 kW fan electrical heater inside. To reduce the complexity of the experiment, a tent was constructed over the test building (see Figure 3b), so that short-wave solar irradiance and wind speed can be neglected on the outside.

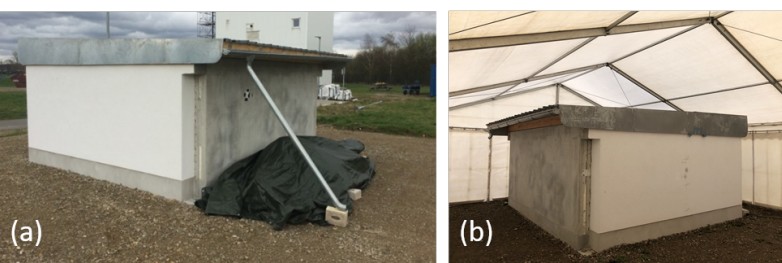

**Figure 3.** The test building without (**a**) and with covering tent (**b**).

Figure 4 shows the setup of the measurement spot. For reference measurements we use two heat flux sensors which are installed directly on the bricks and covered with plaster. Another heat flux sensor is attached to the outer surface of the wall and one on the opposite side of the wall inside of the building. The surface temperatures were also measured with two NTC contact sensors. But their measurements are not used in the following parts of this paper. An additional temperature sensor is mounted on the wall but not attached to the surface, so that it measures the air temperature near the wall surface. Inside the building, another air temperature sensor is installed. Two patches of crumbled aluminium foil are attached to the wall. They serve as diffuse reflectors for long-wave radiation and are used to determine the long-wave radiation coming from surrounding surfaces which is required to calculate the correct wall surface temperature and $T_{surr}$ in Equation (2) [15,34].

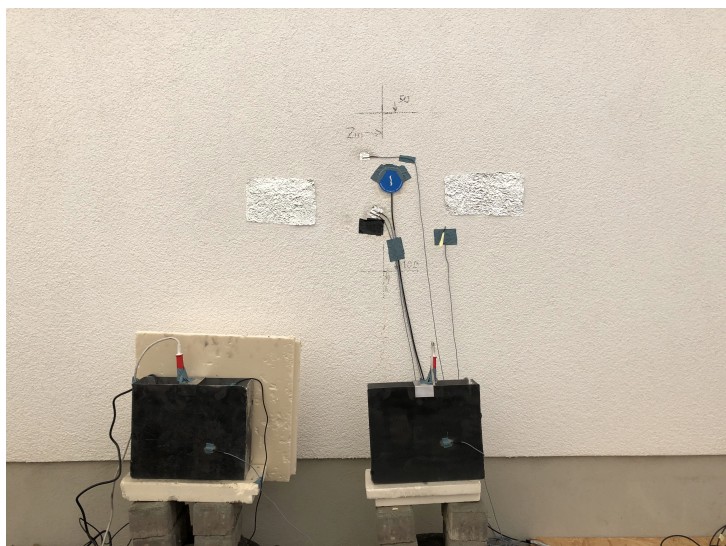

**Figure 4.** The setup of sensors for validation purposes on the measured wall and two blackbodies in front of it. The blackbodies are used to have a better control of the IR camera uncertainty. The patches of aluminium foil serve as a diffuse reflectors for thermal radiation from surrounding objects.

The IR camera was placed approximately five meters away from the wall on a tripod. Its field of view covers the complete scene that is shown in Figure 4. We used a freshly calibrated microbolometer camera with an accuracy of $\pm 1.5$ K and a wave length range from 7.5 to 14 μm. As the IR camera that was used for the experiment did not correct properly for the varying radiation from its housing and the readings changed with the outside air temperature, this correction had to be done in the post processing. In Figure 4 two self-made black bodies can be seen in the bottom. These had different temperatures and served as references for the correction process. We assumed a linear dependence between the housing temperature and the deviation between the measured and real surface temperatures in the relevant temperature range. This allows for estimating a simple linear correction function.

An examplary image of the IR measurement is shown in Figure 5. In the experiment a series of images was taken for five days with a frequency of one image per minute. We chose a region of approximately 25 cm$^2$ in the scene (see Figure 5) on the wall that is located in the center of one of the bricks. Thus, the heat flux is expected to be homogeneous and perpendicular to the wall plane in this area. This also ensures the best comparability with the reference measurements as the heat flux sensors are also placed close to the center of a brick. We used the manufacturer's software to extract the average temperature for this region in every image. Various values are reported in the literature for the emittance of plaster but they are from a rather narrow range [35–37]. For our calculation, we choose $\varepsilon = 0.91$ and for the indoor convective heat transfer coefficient $h_{c,in} = 2.5$ W/m$^2$K as it is recommended in the standard ISO 6946:2017 for mainly horizontal heat flow [38].

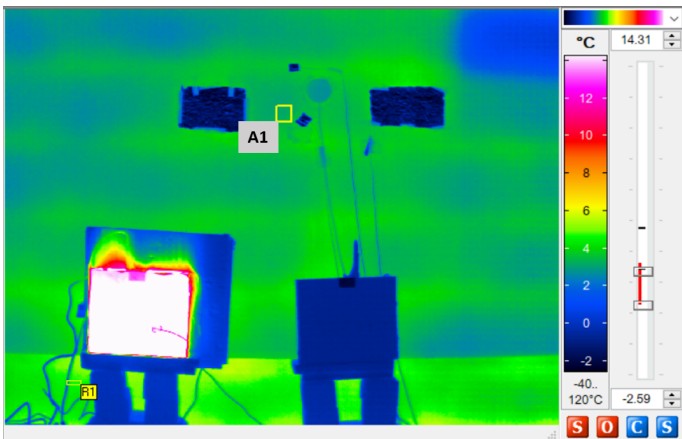

**Figure 5.** An exemplary image from the IR time series. For the model calibration the temperature of a small area that is marked with a yellow square and denoted A1 was used.

## 3. Experimental Results

Figure 5 shows a single IR image from the time series. In the upper right corner of the image a cooler region is visible which is due to a different type of material that is used in the respective region of the test wall. The measurement area A1 is far enough from this region to assume no influence of the lower conductance on the transmittance measurement. On the whole wall the structure of bricks and joints can be seen. The joints appear to have a somewhat higher thermal conductance than the bricks. Figure 6 shows the complete time series of IR temperature measurements in the area A1 together with the outcome of a simulation with the optimized set of resistance and capacitance values. For validation purposes, simultaneous measurements of the NTC and heat flux sensors were taken. Using these values with the standard heat flux meter method [9], a reference thermal transmittance $U_{ref}$ can be calculated from the air temperature and heat flux data. We find $U_{ref} = 1.28 \, W/m^2K$ which corresponds to a thermal resistance of $R_{ref} = 0.098 \, K/W$. The model calibration is used to determine the resistance values of the resistor elements in the model which sum up to the total thermal resistance $R_{tot}$ of the wall. Here, we use the assumption that the wall is symmetric, so that we optimize only three parameters. The calibration is performed for the complete time series of five days and several intervals within this series. The results can be found in Table 1.

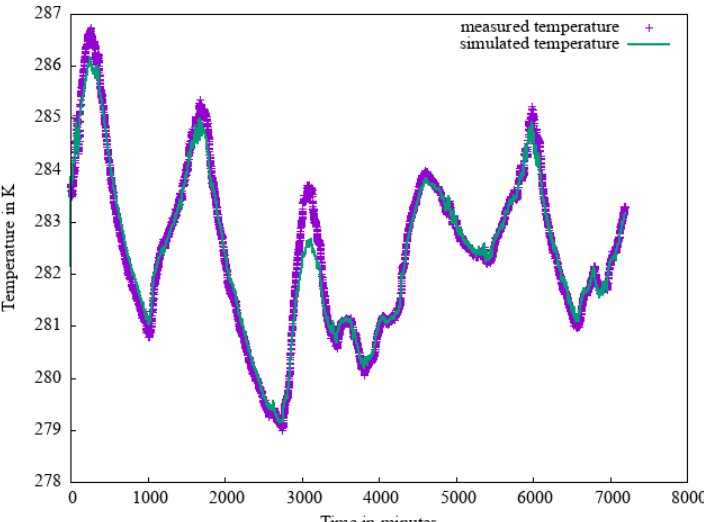

**Figure 6.** Time series of corrected IR camera readings and simulated surface temperatures for a period of five consecutive days for the experiment at the test walls after optimization of wall resistance and capacitance values.

**Table 1.** Results of the model calibration for the thermal resistance and the thermal transmittance using different time periods.

| Days | $R_{\text{tot}}$ [K/W] | $U$ [W/Km$^2$] | Deviation from $R_{\text{ref}}$ [%] | $\chi_r^2$ | $C$ [J/K] |
|---|---|---|---|---|---|
| 1–5 | 0.085 | 1.47 | 13.2 | 0.04 | $1.53 \times 10^6$ |
| 1–3 | 0.083 | 1.51 | 15.4 | 0.04 | $1.39 \times 10^6$ |
| 2–4 | 0.086 | 1.46 | 12.6 | 0.03 | $1.83 \times 10^6$ |
| 3–5 | 0.087 | 1.44 | 11.7 | 0.02 | $1.65 \times 10^6$ |
| 1 | 0.081 | 1.55 | 17.8 | 0.05 | $1.22 \times 10^6$ |
| 2 | 0.085 | 1.47 | 13.3 | 0.01 | $1.88 \times 10^6$ |
| 3 | 0.086 | 1.46 | 12.8 | 0.01 | $1.11 \times 10^6$ |
| 4 | 0.088 | 1.41 | 9.8 | <0.01 | $1.9 \times 10^6$ |

The thermal transmittance determined by the model calibration approach is between 10.1% and 18.0% higher than the reference value ($U_{\text{ref}} = 1.28\,\text{W/m}^2\text{K}$) and between 24.8% and 16.5% higher than the thermal transmittance given by the manufactuerer ($U_{\text{man}} = 1.21\,\text{W/m}^2\text{K}$). A further observation is that small $\chi_r^2$ correlate with small deviations from the reference thermal transmittance. The same observation is made if the manufacturer's value is used as reference. We also observe that the thermal transmittances that come from the model calibration appear to lie systematically higher than the reference value. However, as the reference comes from a single measurement which also carries errors from measurement uncertainties, an uncertainty analysis is performed in Sections 4.1 and 4.2 to find out whether the results of the new method are compatible with the reference.

The HFM method does not give a reference value for the heat capacity. Table 1 shows that the relative variations of the heat capacity are stronger among the different time periods than for the thermal resistance. We cannot identify a clear correlation between the values of $C$ and $\chi_r^2$, either. This indicates that the fit quality is not very sensitive to the value of the heat capacity. We confirmed this by some experiments where we kept the heat capacities constant during the fit and found a weak dependence of the resulting thermal resistance on the chosen capacities.

## 4. Uncertainty Analysis

In order to assess whether the results of our new method and those of the reference method are compatible, an uncertainty analysis was conducted for both methods. Overlapping conficence intervals will indicate that the results are compatible.

### 4.1. Uncertainty Analysis of the HFM Method

ISO 9869-1:2015 [9] explains how much uncertainty should be considered for each parameter in the HFM method. It depends on the following factors,

- Approx. 5% error due to the accuracy of the calibration of the HFM and temperature sensors if the sensors are well calibrated.
- Approx. 5% variation due to slight difference in thermal contact between HF sensor and the wall surface.
- 2% to 3% uncertainty due to operational error of the HFM.
- Approx. 10% error caused by the variations over time of the temperatures and heat flow.
- Approx. 5% error in thermal transmittance measurement due to temperature variations within the space and difference between air and radiant temperatures.

These above uncertainty factors are added to produce the total uncertainty. The norm ISO 9869-1:2015 says that if the suggested measurement conditions are fulfilled, the total uncertainty shall be expected to be between the quadrature sum and arithmetic sum [9]. As a result, it stays between 14% to 28%. As the thermal transmittance that is measured with the HFM method is $U_{\text{ref}} = 1.28\,\text{W/m}^2\text{K}$ the confidence interval ranges from $1.1\,\text{W/m}^2\text{K}$ to $1.46\,\text{W/m}^2\text{K}$ in the best case and from $0.92\,\text{W/m}^2\text{K}$ to $1.64\,\text{W/m}^2\text{K}$ in the worst case. The

observed deviations in Table 1 are, therefore, within the expected range of uncertainty of the reference method.

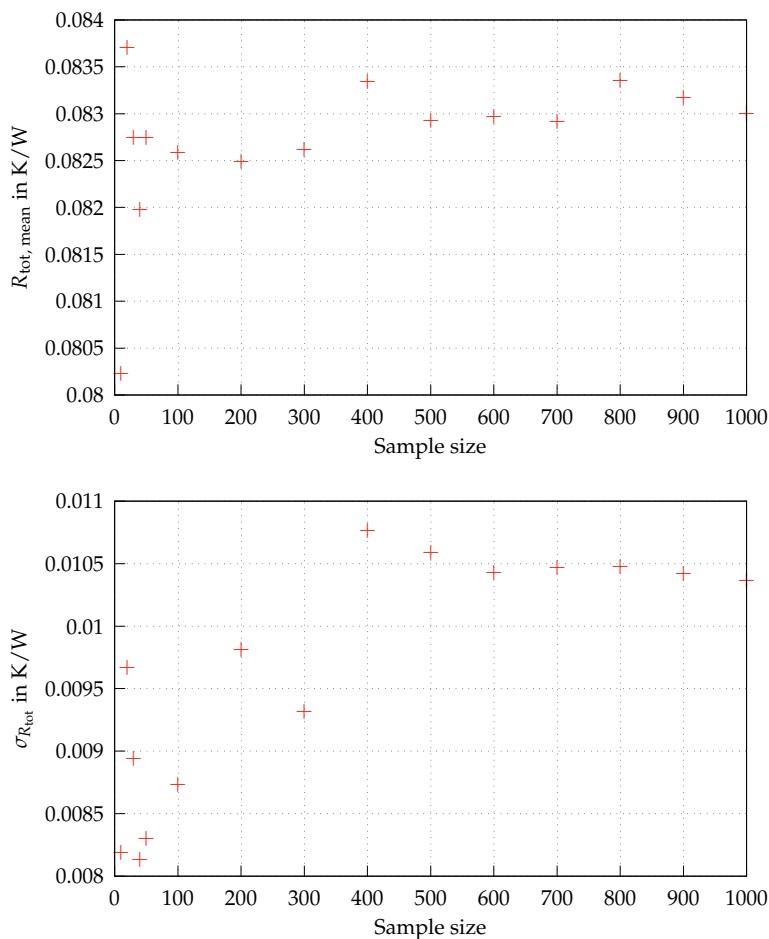

**Figure 7.** Mean value (**top**) and standard deviation (**bottom**) of the thermal resistance for sample sizes of 10 to 1000.

### 4.2. Uncertainty Analysis of the Model Calibration Approach

To determine the error of the thermal resistance obtained by the calibration method, we have to estimate the error in the input quantities. For the quantities that we measure directly or indirectly we rely on information about measurement errors from the manufacturers. The error in the emittance $\epsilon$ is estimated from the range of different values for plaster in the literature [35–37]. The error in the internal convective heat transfer coefficient is based on an ISO standard [38]. We assume a rather large error here to account for the fact that the indoor conditions are not observed in our scenario. The errors are shown in Table 2. We perform a latin hypercube sampling of the values assuming a normal distribution with the given error as standard deviation. Each sample is a different set of values for the input quantities. For each sample the calibration is performed using the complete time series of five days. We set a threshold of $\chi_r^2 = 0.1$ to filter out samples that clearly show a bad fit and determine the mean and standard deviation of the remaining ones. We vary the number of samples from 10 to 1000 to check for what sample size the mean and standard deviation of the thermal resistance converges. Figure 7 shows that for >500 samples we can consider the process as converged. So we find that the converged mean total thermal resistance is $R_{\mathrm{tot,conv}} \approx 0.083\,\mathrm{K/W}$ and the corresponding standard deviation is $\sigma_{\mathrm{tot,conv}} \approx 0.01\,\mathrm{K/W}$ which corresponds to an error of $\sim 12\%$. This means that the reference value determined by the HFM method $R_{\mathrm{ref}} = 0.098\,K/W$ lies within the $2\sigma_{\mathrm{tot,conv}}$ interval of the result that we obtained with our calibration method.

**Table 2.** The errors of the input quantities for the model calibration. The second column shows the source of the signal.

| Quantity | Source | Error |
|:---:|:---:|:---:|
| Outside $T_{air}$ | NTC Sensor | $\pm 0.22$ K |
| Outside $T_{wall}$ | IR camera | $\pm 1.50$ K |
| Outside $T_{surr}$ | IR camera on aliminium foil | $\pm 1.50$ K |
| External $h$ | Nusselt number correlation | $\pm 0.50$ W/m$^2$K |
| $\varepsilon$ | Literature [35–37] | $\pm 0.03$ |
| Inside $T_{air}$ | NTC Sensor | $\pm 0.22$ K |
| Inside $T_{surr}$ | NTC Sensor | $\pm 0.22$ K |
| Internal $h$ | Literature [38] | $\pm 3.00$ W/m$^2$K |

## 5. Discussion

This paper presents a model calibration approach to determine the thermal resistance, capacitance and, consequently, the thermal transmittance of building wall elements from remote surface temperature measurements with an IR camera. The wall element is modelled with a lumped R-C-type model and is calibrated by minimizing an objective function that compares the measured and simulated external surface temperature of the element. We reported an experiment using infrared cameras to test the method on a real wall and compare the outcome to a standardized reference HFM measurement. The measurement data gives very consistent results of the thermal resistance for various time intervals ranging from a single day to five days, while standard methods usually require measurement periods of up to seven days [9]. The large variation in the heat capacity among the different intervals indicates that the approach is less well suited to accurately determine this quantity for a building wall. This is consistent with the work of Arregi et al. [25] who find that lumped resistance-capacitance models are not well suited to determine the heat capacitance of a concrete wall. We have performed an uncertainty analysis of our transmittance measurement for the same dataset and a standard reference measurement that uses HFM. It turns out that the confidence intervals overlap which indicates that the new method achieves a satisfactory accuracy. The reduced chi-square of the model fit appears to be a good indicator for the success of the calibration process and the accuracy of the resulting thermal transmittance. High $\chi_r^2$ may indicate large measurement errors or inadequate assumptions for input quantities.

The experiment was performed under simplified conditions: Short-wave solar radiation and wind speed could be assumed to be zero because the test building and the surrounding area were covered with a tent. While the solar irradiation onto the wall can be easily measured and modelled it may lead to rapid changes in the temperature of the wall close to the outer surface. Such effects may require wall models with more than two thermal masses and three resistors. The wind, on the other hand, is less likely to cause such surface-effects. However, the wind speed varies significantly in front of large walls, so that a measurement in one or few locations may not be representative for the complete surface. The model that is chosen to connect wind speed and the surface heat transfer coefficient will also change the calibration outcome. So further tests under full environmental conditions are necessary to determine the conditions under which our method reliably gives accurate results for the thermal transmittance.

In the experiment that we reported we used the temperature average of a single relatively small area to perform the model calibration. The same procedure can be run for additional areas on the wall to obtain the thermal resistance and thermal transmittance with a spatial resolution on the wall. The size and the number of the averaging areas must be chosen according to the respective measurement task. Small areas will likely introduce some noise in the resistances because noise coming from the camera measurements is not averaged out. If the area is large it increases the chance that inhomogeneities are missed.

It is desirable to use the method also for other components of building envelopes than in our example. The necessary adaptations mainly depend on the reflective properties

of the surface material. Most usual plasters, bare bricks, wooden surfaces, dull tiles, or many plastic materials will only require a change of the emittance for long-wave radiation and the absorptance for short-wave radiation in the model. Metallic surfaces often have high reflectance in the long-wave range. As long as diffuse reflection clearly dominates over specular reflection no change in the methodology or setup is required. However, a strong contribution of reflected thermal radiation to the measurement by the IR camera may impact the accuracy. This may be compensated by extending the measurement time and including mostly measurement points which were taken during the night in the calibration. However, this scenario has not been examined experimentally, so far, and will be subject of future work. Materials that reflect the long-wave thermal radiation specularly, e.g., glass, must be treated in a different way to subtract the signal of any hot object being reflected on the measurement surface. Different approaches will be tested in future. Roofs may pose an additional challenge if the space inside is not heated. First of all, the information whether the space below the roof is heated or not has to be obtained from additional sources, such as building plans, or the inhabitant's knowledge. For unheated spaces, a smaller indoor temperature must be assumed which will probably increase the necessary measurement time to achieve the same level of accuracy.

## 6. Summary

To conclude, the presented method offers an efficient way to determine the thermal transmittances of a building wall. An experiment with an uncertainty analysis on a real wall under simplified environmental conditions indicates that its accuracy is at least comparable to the reference heat flux meter method. Further tests and validation steps must be undertaken to ensure the reliability of the method under full environmental conditions and for surface materials with reflective properties that differ significantly from those of the plaster in our experiment.

The main advantage of this method is the fact that the measurements can be done completely from the outside of the building and no measurement equipment has to be installed on the building surfaces. This implies minimal impact on the usage and operation of the building. Compared to state-of-the-art methods that rely on the stationarity assumption, our method also shows the potential to reduce the required measurement times. However, as the method relies on visual access to the respective envelope surfaces, it may not be possible to study all parts of the building envelope—especially for buildings with a complex structure. In addition, for high buildings the angle dependence of the thermal emittance may become important which is not considered by the method, so far. We believe that due to its advantages, the presented method has a good chance to, nevertheless, turn out as a valuable addition to the state of the art.

**Author Contributions:** Conceptualization, J.E.S. and M.R.; software, D.P.; validation, D.P.; writing—original draft preparation, J.E.S.; writing—review and editing, D.P. and M.R.; supervision, B.H.; project administration, J.E.S. and M.R.; funding acquisition, J.E.S. All authors have read and agreed to the published version of the manuscript.

**Funding:** This research was funded by the German Federal Ministry for Economic Affairs and Climate Action.

**Institutional Review Board Statement:** Not applicable.

**Informed Consent Statement:** Not applicable.

**Data Availability Statement:** The data that is required to understand the method and experiments presented in this paper are is given in the text. Additional data, e.g., raw measurement data can be provided upon request by the authors.

**Acknowledgments:** The authors would like to thank Johannes Pernpeintner for his support with the experiment and helpful discussions of the methodology.

**Conflicts of Interest:** The authors declare no conflict of interest. The funders had no role in the design of the study; in the collection, analyses, or interpretation of data; in the writing of the manuscript; or in the decision to publish the results.

## Abbreviations

The following abbreviations are used in this manuscript:

| | |
|---|---|
| C | capacitor |
| EU | European Union |
| HFM | Heat flux meter |
| LW | long wave |
| HVAC | Heating, ventilation, air-conditioning, and cooling |
| IR | Infrared |
| R | resistance |
| SW | short wave |
| TM | Thermal mass |

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
