# Peer review of "A Model Calibration Approach to U-Value Measurements with Thermography"

_buildings, doi:10.3390/buildings13092253_

Round 1
Reviewer 1 Report
The authors developed a technique, based on a simplified electrical analogy building envelope model that can simulate the buildings dynamic thermal behavior. The model’s tunable parameters are fitted based on the infrared measurement of the building’s surface temperatures. The result is a simple method of surveying the thermal properties of the building envelope (U-values and capacitance).
The method is promising, as shown by the small demonstration included. However, it needs further refinement.
The authors are advised to further improve their paper regarding the following points, in order to better explain and support their methodology.
Please refer to “U-value” as “thermal transmittance”.
Figure 2: The selection of three resistors for the wall model suggests a “sandwich” construction with the insulation material between two wall layers? Please discuss. Your wall construction is different.
Line 196: However, ISO 6946 would suggest Rsi=0.13 which is 7.69 W/m2K, not 2.5 W/m2K!
Figure 5: why you selected area A1 as reference? Please discuss.
Figure 5: Does your camera have selectable a more narrow range, say between 0 -55 oC?
Figure 5: the specific texture of the stones is visible through the infrared picture? Please explain. Which is the binding material and its thermal properties? Do they interfere with the stone properties in producing the increased apparent transmittance?
Figure 7 increase resolution.
Lines 306-307 have you tried to repeat the measurements during the night?
The limitations of the methodology should be further discussed by the authors.
The literature survey needs further expansion. This is a popular research subject.
Some minor English editing will be required
Reviewer 2 Report
The paper is well-written and the study is fascinating. I do not have any revision comments and suggest to accept it as it is.
Reviewer 3 Report
The paper deals with an important issue related to the assessment of the actual thermal quality of building envelope, however, some elements need to be corrected before being considered for publication.
The article raises an important problem related to the assessment of the thermal quality of building envelope, but some elements require correction before being considered for publication.
Introduction:
- The sentence in lines 16-17 “In most climate zones heating and cooling are responsible for the largest portion of energy consumption during the operation of a building” is very general and needs to be clarified - what specific climates are meant in individual cases? In which climate is it most important and in which is it least important - this would help to highlight the usefulness and importance of the subject of the article in individual countries.
- In line 26 should it not be added: by 1 m2 of the envelope element? Or by the area unit of the envelope element?
- Lines 29 and 32 - what exactly follows from the cited references - what are the differences in U-values measured on real buildings and what can be the gap in energy efficiency between simulation and real energy consumption? Each reference should be properly subsumed. Similarly in line 34 - how capacity can flatten this demand?
- Line 42 - the abbreviation IR should be placed after “Infrared thermography”.
- Lines 53: It should be stated what follows from references 14 and 15 - how sensitive are these results to changing external environment conditions? This would be an interesting background to the question of paper.
- The main results of references 16-18 (line 62) should be described in the paper
- Check if writing in the third person "we...", “our method”, e.t.c, is accepted by the journal?
Method
- The first sentence of the "Method" section is not very clear
- The symbols in figure 1 are not explained next to the figure, they are not explained until much further. There is no information whether it is an own development or requires confirmation of copyright or references.
- Does the method apply to all elements of the building envelope or only to the walls? If it applies only to walls, it should be indicated throughout the article (in the title and in the following). If it also concerns other elements of the building envelope (e.g. roof), appropriate drawings are necessary - similar as for walls, as there will be some differences. If you mean not only walls, then also symbol descriptions from formulas (e.g. “the right and left side ...” – line 94) are not correct
- Explain the abbreviations used on line 120
- There is no source in Fig. 2
- Refer to formula numbers in the text
- Temperature and derivative with respect to time, should not both be capitalized T
- line 97 - explanations of the red lines probably refer to figure 2, so they should be closer to it, not formula 2. Are you sure that the temperature at different points in the cross-section of the envelope can be equal?
- The assumption that the wall is symmetrical is a great simplification and very often it does not match the real construction of walls and other types of partitions. It should be explained and described why it was assumed so and, consequently, narrow down the sample for which the model can be used.
- line 107: The assumption that the wall is symmetrical is a great simplification and very often it does not match the real construction of walls and other types of partitions. It should be explained and described why it was assumed so and, consequently, narrow down the sample for which the model can be used.
- What is the difference between Tair (in formula 3) and T1,r (in formula 1) and Tsurr (in formula 4). In the opinion of the reviewer, other symbols should not be used for the same physical quantities.
Experimental Results
- The first paragraph of section 3 contains input data that should be in the previous chapter, not in the "Results" section.
- Line 196 - why the indoor convective heat transfer coefficient equal 2.5 was adopted?
- In subsection 3.1, which is part of section 3 (results), there is again a description of the method. The structure of the article should be improved.
- On the basis of the discrepancies in the results, which are significant, the practical sense of such an approach should be clarified.
- Are you sure Fig. 7 shows the thermal resistance?
Discussion
- Section 4 should have a different title because it contains summary and conclusion elements, not discussion.
- It should also be emphasized what is the innovation of this study compared to others and the possibility of practical application of the proposed method.

Round 2
Reviewer 1 Report
Accept
Author Response
Dear reviewer,
thank you for your time and the evaluation of our work.
Kind regards,
Dhruvkumar Patel, Jacob Estevam Schmiedt, Marc Röger, Bernhard Hoffschmidt
Reviewer 3 Report
Thank you for revising your work. The authors have successfully considered the suggestions of the reviewer.
I had one further comment that I would like the authors to explain. This should not require significant changes to be made to the paper, but I think it is necessary.
Because ISO 6946 suggest Rsi=0.13 which is 7.69 W/m2K, not 2.5 W/m2K in the case of the type of partition presented by the authors, it is necessary to add information on how the results of the analyzes changed with the correct value of the surface resistance.
After completing this comment on the result related to the change in surface resistance, the manuscript should be considered for publication in the journal.
Author Response
Dear reviewer,
thank you for your comment. The recommendation of ISO 6946 is to use h = 7.69 m2K/W for the heat transfer coefficient that combines the convective and the (linearized) radiative heat exchange with the environment. As we included the full radiative exchange with the Stefan-Boltzmann equation in our model we use h = 2.5 m2K/W as it is recommended by ISO 6946:2017 in table C.1. We made some changes to the text in lines 180 - 184 to clarify that we model the convective and radiative heat exchange separately on the inner and outer side of the wall.
Kind regards,
Dhruvkumar Patel, Jacob Estevam Schmiedt, Marc Röger, Bernhard Hoffschmidt